# The Changing Therapeutic Landscape of Metastatic Renal Cancer

**DOI:** 10.3390/cancers11091227

**Published:** 2019-08-22

**Authors:** Javier C. Angulo, Oleg Shapiro

**Affiliations:** 1Departamento Clínico, Facultad de Ciencias Biomédicas, Universidad Europea de Madrid, Hospital Universitario de Getafe, Carretera de Toledo km 12.5, Getafe, 28043 Madrid, Spain; 2SUNY Upstate Medical University, Upstate University Hospital, Syracuse, NY 13210, USA

**Keywords:** renal cell carcinoma, immune checkpoint inhibitors, tyrosine kinase inhibitors, efficacy, toxicity, cytoreductive nephrectomy

## Abstract

The practising clinician treating a patient with metastatic clear cell renal cell carcinoma (CCRCC) faces a difficult task of choosing the most appropriate therapeutic regimen in a rapidly developing field with recommendations derived from clinical trials. NCCN guidelines for kidney cancer initiated a major shift in risk categorization and now include emerging treatments in the neoadjuvant setting. Updates of European Association of Urology clinical guidelines also include immune checkpoint inhibition as the first-line treatment. Randomized trials have demonstrated a survival benefit for ipilimumab and nivolumab combination in the intermediate and poor-risk group, while pembrolizumab plus axitinib combination is recommended not only for unfavorable disease but also for patients who fit the favorable risk category. Currently vascular endothelial growth factor (VEGF) targeted therapy based on tyrosine kinase inhibitors (TKI), sunitinib and pazopanib is the alternative regimen for patients who cannot tolerate immune checkpoint inhibitors (ICI). Cabozantinib remains a valid alternative option for the intermediate and high-risk group. For previously treated patients with TKI with progression, nivolumab, cabozantinib, axitinib, or the combination of ipilimumab and nivolumab appear the most plausible alternatives. For patients previously treated with ICI, any VEGF-targeted therapy, not previously used in combination with ICI therapy, seems to be a valid option, although the strength of this recommendation is weak. The indication for cytoreductive nephrectomy (CN) is also changing. Neoadjuvant systemic therapy does not add perioperative morbidity and can help identify non-responders, avoiding unnecessary surgery. However, the role of CN should be investigated under the light of new immunotherapeutic interventions. Also, markers of response to ICI need to be identified before the optimal selection of therapy could be determined for a particular patient.

## 1. Immune Checkpoint Inhibition in Renal Cancer

The ability to evade immune surveillance and programmed cell death characterize kidney cancer cells. Some tumors express biomarkers to prevent or elude an immune response, which is crucial in not allowing cells with damaged genetic load to proliferate. Cellular damage causes cell division arrest, so the cell can repair itself, and cell death is induced if repair is not possible to avoid the development of a malignant cell line. Restoring the ability of the immune system to function through its various checkpoints is mandatory. In this scenario, T-regulatory cells play a significant role in regulating the immune response to what the body recognizes as foreign [1,2]. Targeting immune checkpoints in clear cell renal cell carcinoma (CCRCC) is being extensively analyzed currently [3,4,5]. The pitfalls of the clinical translation of PD-1/PD-L1 blockade have also been critically reviewed [6,7,8].

CTLA-4 (CD15) is found on T-cells and if activated, results in the inhibition of T-cell function. Ipilimumab, investigated in patients who previously received IL-2, induced autoimmune events. Of patients with a sustained response, 30% had an autoimmune event [9]. Tremelimumab has also been evaluated in patients with metastatic (CRRCC) in association with sunitinib, a vascular endothelial growth factor (VEGF) inhibitor, and durable response was confirmed in 43% of the cases [10].

PD-1 (PDCD1) is a type I transmembrane glycoprotein receptor, part of the CD28/CTLA-4 immune checkpoint receptor family, expressed on peripheral blood mononuclear cells and activated tumor-infiltrating mononuclear immune cells and responsible for the down-regulation of T-cells. PD-1 is monomeric and contains a single immunoglobulin-like variable (IgV) domain in its N-terminal extracellular region, which mediates PD-1 binding to its ligands PD-L1 and PD-L2 [11,12]. The PD-1 intracellular region contains two immunoreceptor tyrosine-based regulatory structures that experience tyrosine phosphorylation and are responsible for the binding to the SH2-domain-containing tyrosine phosphatases PTPN6 (SHP1) and PTPN11 (SHP2), thus inhibiting T and B cell antigen receptor-mediated signaling [13]. A serum soluble variant of PD-1 has been found, although its relevance in CCRCC remains to be determined [8].

PD-1 ligand PD-L1 (CD274) is another type I transmembrane glycoprotein, part of the B7 family of immune checkpoint proteins [14]. PD-L1 expression correlates with VHL inactivation and HIF-2α expression [15,16], and carries bad prognosis for patients with CCRCC [17,18]. PD-L2 (PDCD1LG2) is another PD-1 ligand, a closely related protein to PD-L1. Both PD-1 ligands are expressed in kidney epithelial cells under normal conditions and upregulated by inflammation [19]. A serum soluble PD-L1 associated with tumor aggressiveness has been detected in patients with CCRCC [20]. 

Ipilimumab and nivolumab are monoclonal antibodies targeting the immune checkpoint proteins CTLA-4 and PD-1, respectively. PD-1 acts as a negative regulator of T-cell activity by binding to PD-L1 on either antigen-presenting or tumor cells, causing the inhibition of T-cell anti-neoplastic responses. CTLA-4 acts as a negative regulator of T-cell activation by binding to the B7 ligand CD80, and CD86 expressed on antigen-presenting cells, thus preventing the interaction between CD28 and the B7 ligands. Nivolumab binds to PD-1 and blocks the inhibitory signaling of the PD-1/PD-L1 interaction. Ipilimumab binds to CTLA-4 and blocks the inhibitory signaling of the CTLA-4/B7 interaction (Figure 1).

Nivolumab was approved by the FDA in 2015 as second-line therapy for mCCRCC after the results of the Phase 3 Checkmate-025 trial (NCT01668784) were published. The trial revealed superiority in overall survival in the nivolumab group compared to the everolimus group. Shortly thereafter, treatment with nivolumab monotherapy became the standard of care for patients who progressed after initial treatment with a VEGF inhibitor. After the Checkmate-214 trial (NCT02231749) results came to light in 2018, the combination of nivolumab plus ipilimumab was approved as first-line therapy for intermediate and poor-risk patients [21], thus totally changing the therapeutic landscape of advanced CCRCC. A better knowledge of the immunology of T-cell activation is leading to the establishment of immune checkpoint inhibition (ICI), and the beginning of a new era in the treatment paradigm of patients with advanced CCRCC, using monoclonal antibodies to block the inhibition of T-cell activation.

## 2. The New Paradigm to Treat Metastatic Renal Cancer

Systemic therapy is the mainstay of treatment in patients with mCRRCC. The last 15 years saw a revolution in therapy based on VEGF-inhibition and immune checkpoint inhibition (ICI). Trials have shown a durable response in patients and an increase in the overall survival. The drastic change in the treatment paradigm happened in 2007 once the tyrosine kinase inhibitor (TKI), sunitinib, a potent VEGF inhibitor, proved superior to interferon-alfa in the treatment of metastatic clear cell renal cell carcinoma [22,23]. In 2015, another important trial showed nivolumab, the programmed cell death 1 (PD-1) receptor inhibitory signal blocker, to be superior to everolimus. This allowed immunotherapy to become the standard of care as second-line therapy for metastatic CCRCC [24]. 

Advances in risk group stratification were a major catalyst needed for the evolution of treatments and better interpretation of trial data. Original risk group categories were proposed by the Memorial Sloan Kettering Cancer Center (MSKCC) in the era of interferon-alpha and were widely used until very recently. This classification consists of five prognosis predicting factors, including time from the initial diagnosis to the start of systemic therapy, Karnofsky performance status, hemoglobin, serum calcium, and lactate dehydrogenase [25]. A similar classification system was proposed by Heng et al., taking into account neutrophil and platelet counts [26]. Accumulation of risk factors define favorable (0 positive factors), intermediate (1–2 factors), and high-risk (3 or more factors) groups. The International Metastatic Renal Cell Carcinoma Database Consortium (IMDC) validated Heng’s criteria in patients treated with first, or second-line VEGF targeted therapy, making it applicable to a more contemporary cohort of patients [27].

The Checkmate-214 trial (NCT02231749) looked at the combination nivolumab plus imilimumab versus sunitinib in mCRRCC. The risk categories were defined as “good” (favorable) and “bad” (intermediate/poor) in the trial. The results revealed that in the metastatic setting, the combination of nivolumab plus ipilimumab demonstrated obvious superiority in the treatment-naïve patients with intermediate and poor-risk mCRRCC (objective response rate (ORR) 42% vs. 29%; *p* < 0.0001). Interestingly, the combination of nivolumab plus ipilimumab did not demonstrate superiority for favorable-risk disease (ORR 39% vs. 50%; *p* = 0.14). Paradoxically, there was a noticeable trend towards improvement in the progression-free survival with sunitinib versus the combination therapy (25.1% vs. 15.3%; *p* < 0.0001) [21,28,29]. Subsequently, the National Comprehensive Cancer Network (NCCN) Guidelines for metastatic kidney cancer have adopted the combination of ipilimumab plus nivolumab as the first-line therapy in the intermediate and poor-risk groups [30]. The tolerability of this combination immunotherapy was acceptable, despite the fact that more patients discontinued the therapy as compared to the sunitinib arm (24% vs. 12%). The most frequently seen grade 3–4 immune-related adverse effects (AEs) were diarrhea, hepatitis, and hypophysitis. Almost 60% of the patients with AEs required corticosteroids to manage their symptoms [21,29].

Further, the shift in first-line management of metastatic RCC has occurred as the results of the Keynote-426 trial (NCT02853331) became available. Pembrolizumab plus axitinib were shown to be superior to sunitinib regardless of the risk groups (ORR 59% vs. 35%; *p* < 0.001), with an acceptable safety profile [31]. Furthermore, the Javelin Renal-101 trial (NCT02684006) revealed avelumab plus axitinib to be more efficacious than sunitinib (ORR 51% vs. 25%). The Hazard Ratio (HR) for progression to death was 0.50 (95% CI 0.26–0.97) for favorable, 0.64 (0.47–0.88) for intermediate and 0.53 (0.30–0.93) for poor International Metastatic Renal Cell Carcinoma Database Consortium (IMDC) risk groups [32]. These trials cemented the strategy of using combined immune checkpoint and VEGF inhibition in patients with previously untreated metastatic CRRCC. This treatment paradigm has found its way into the NCCN and European Urological Association guidelines (Table 1) [30,33].

The IMmotion151 trial (NCT02420821) evaluated the programmed death-ligand 1 (PD-L1) blocker atezolizumab and VEGF-A inhibitor bevacizumab, compared to sunitinib as first-line therapy. Interim analysis has confirmed the combination of monoclonal antibodies prolonged progression-free survival in the PD-L1 positive patients (HR = 0.74; 95% CI 0.57–0.96; *p* = 0·02), but not in the overall population [34]. Two additional Phase III trials investigating different combination strategies, such as cabozantinib plus nivolumab compared to sunitinib (Checkmate-9ER, NCT01984242), and lenvatinib plus pembrolizumab compared to lenvatinib plus everolimus or sunitinib (Clear, NCT02811861) have not matured as of yet (Table 2).

Combination treatments have shown improved response rates comparing to single-agent therapy with sunitinib and have replaced VEGF-targeted therapy as the standard first-line treatment in good and intermediate-risk groups. Interestingly, combination therapies have replaced the mammalian targets of rapamycin (mTOR) inhibitors, such as temsirolimus and everolimus, which were used for treatment-naïve poor-risk patients and patients treated with VEGF-TKI agents, respectively. However, the toxicity of newer treatment strategies using ICI should be carefully balanced to that of monotherapies. Treating physicians and investigators should take into consideration the incidence of treatment-related grade 3–4 adverse events (AEs) and treatment discontinuation due to these events, before optimal individualized therapy for mCRRCC is decided.

The overall toxicity profile of ICI differs from that of traditional therapies, and a better understanding of the AEs and their optimal management is critical for practising physicians [35]. A systematic review revealed 80% of patients receiving ICI, experienced AE patients with grade 3–4 AEs constitute 20% of the cohort, and less than 10% have to discontinue treatment due to adverse events [36]. Immune-related AEs (irAEs) are due to treatment and most commonly affect the skin (rash, pruritus) 30%, liver (elevated AST and ALT) 20%, gastrointestinal tract (diarrhea) 15%, endocrine system (hypothyroidism) 12%, kidneys (elevated creatinine) 7%, and lungs (pneumonitis) (5%). The most common grade 3–4 irAEs involve the liver. Interestingly, there were no deaths due to AEs reported in the trials reviewed [36].

Most trials and pooled analysis of ICI therapy suggest irAEs may occur anytime from weeks to years after the start of therapy, even after therapy cessation. However, the majority take place within the first year of treatment and resolve with the appropriate therapy [37]. Systemic corticosteroids are the mainstay of treatment for immune complications, but anti-TNF-α can also be used for refractory irAEs [38]. The use of systemic immunosuppressants does not seem to negatively impact the therapeutic effects of ICI therapy [36,37].

Patients who cannot tolerate ICI therapy can alternatively receive VEGF-TKI-based therapy. In these patients, sunitinib and pazopanib appear to be the optimal regimen in the favorable group, and cabozantinib remains a valid option for the intermediate and high-risk groups. However, as ICI is increasingly utilized as the front-line therapy for mCCRCC, limited data exist on the response rates and survival of patients treated with second-line VEGFR-TKI-based therapy. Antitumor activity and tolerance of TKI monotherapy after failed ICI seems comparable to historical data for the first-line TKI regimen [39].

## 3. Current Role of Multikinase Inhibitor Monotherapy

The introduction of VEGF receptor inhibitors, sorafenib and sunitinib in 2005 started a revolution in the management of mCCRCC. These therapies produced response rates of 40% in the front-line setting and progression-free survival estimates in the range of 9 and 12 months [40,41]. Salvage therapy involved treatment with mTOR inhibitor everolimus, but the response rate with this intervention was only modest [42], this created a void in the salvage therapy space. This led to the evolution of other salvage regimens such as multitargeted kinases and immunotherapy [29].

Cabozantinib, a multikinase inhibitor, was approved by the FDA in 2016 for patients with advanced kidney cancer that were formerly treated with one or more antiangiogenic drugs. The drug is a potent inhibitor of MET and VEGF receptor 2, but also of other receptor tyrosine kinases (RET, KIT, AXL and FLT3) [43,44]. It was the first medication that showed a statistical improvement in the three endpoints of clinical efficacy: response rate, progression-free survival, and overall survival. Meteor trial (NCT01865747), which ran concurrently with Checkmate-025, compared cabozantinib and everolimus. Cabozantinib improved progression-free survival (HR 0.51, 95% CI 0.41–0.62) and ORR (17% vs. 3%). The median overall survival was 21.4 months for cabozantinib versus 16.5 months for everolimus (HR 0.66; 95% CI 0.53–0.83). Grade 3–4 AEs occurred in 39% of the cabozantinib group and 40% of patients treated with everolimus. Most common grade 3–4 AEs were hypertension 15%, diarrhea 13%, fatigue 11%, hand-foot syndrome 8%, anemia 6%, and hypomagnesemia 5%. The dose reduction is effective to manage toxicities in this patient population and was required in 60% of the affected cohort in the Meteor trial [45,46,47]. The Cabosun trial (NCT01835158) compared cabozantinib to sunitinib. Unlike Checkmate-214 trial, no patients in the good-risk group by the IMDC criteria were included. Progression-free survival was 8.6 months for cabozantinib and 5.3 months for sunitinib (HR 0.66; 95% CI 0.46–0.95). The overall survival was higher with cabozantinib (30.3 vs. 21.8 months), but the difference did not reach statistical significance (HR 0.80; 95% CI 0.50-1.26) [48]. Since cabozantinib and nivolumab were developed in the same timeframe, there are no studies looking at the optimal sequencing of these agents. The current dogma tells us that patients who have prolonged clinical benefit with initial anti-VEGF therapy and demonstrated tolerability to this therapy are likely to benefit from cabozantinib as second-line treatment at progression [49]. Still, real-world data indicate comparable overall survival and time to treatment failure for nivolumab and cabozantinib. Therefore, both are reasonable therapeutic options in patients experiencing progression after initial first-line VEGF-TKI agents [50].

The Axis trial (NCT00678392) compared the efficacy and safety of axitinib versus sorafenib as second-line treatment. The overall survival did not differ between the two groups, but the progression-free survival was longer for axitinib (HR 0.656; 95% CI 0.552–0.779). Common grade 3–4 AEs were hypertension (17%), diarrhea (11%) and fatigue (10%) in axitinib-treated patients and hand-foot syndrome (17%), hypertension (12%) and diarrhea (8%) in sorafenib-treated patients [51]. These data allowed axitinib to become another second-line treatment option after first-line TKIs sunitinib, sorafenib, or pazopanib [52]. Optimal sequence and selection of nivolumab, cabozantinib, and axitinib remain undefined [53]. The reimbursement landscape differs around the world and often limits treatment options [54].

## 4. Cytoreductive Nephrectomy in the Era of Immunotherapy 

Based on retrospective data, traditional treatment of mCCRCC includes a combination of VEGF-TKI-targeted therapy and cytoreductive nephrectomy (CN). This approach has recently become a matter of debate as new data suggest the lack of survival benefit for patients undergoing CN. A recent meta-analysis evaluating the efficacy and safety of perioperative sunitinib in patients with metastatic and advanced renal cancer revealed superior response rate, overall survival, and progression-free survival [55]. The randomized controlled study, Carmena (NCT00930033), has failed to show that CN plus sunitinib is superior to sunitinib alone in terms of overall survival (HR 0.89; 95% CI 0.71–1.10). Non-inferiority of targeted therapy alone was demonstrated. Also, CN was associated with a significant risk of perioperative mortality and morbidity. However, among many limitations of this study was the selection of many poor-risk patients for cytoreductive nephrectomy, who were unlikely to benefit from surgical intervention anyway. Based on these results, CN should be re-considered in many poor and intermediate-risk patients. Most good-risk patients would still likely benefit from cytoreductive nephrectomy [56,57].

Surtime (NCT01099423) compared immediate surgery versus neoadjuvant sunitinib followed by surgery. The progression-free rate at 28 weeks was not improved in patients treated with neoadjuvant sunitinib (43% vs. 42%; *p* = 0.61); however, more patients received sunitinib, and CN could be avoided in those with progressive disease [58]. In summary, neoadjuvant sunitinib may identify patients who are non-responders to systemic therapy, in whom CN could be safely avoided without affecting the outcome. Conversely, a minimally invasive approach and sometimes nephron-sparing surgery could be performed in selected patients [59,60].

As stated above, the superiority of nivolumab and ipilimumab over sunitinib has led to a paradigm shift in the first-line treatment of intermediate and poor-risk patients. Unfortunately, the role of CN in the setting of a novel immunotherapy is unknown and should be investigated [61]. One out of five patients entering Checkmate-214 and demonstrating a survival benefit with ICI had their primary tumor in place. That means the role of CN needs to be better defined in the era of immunotherapy.

## 5. The Need for New Markers in the Era of Immunotherapy

Treatment options for mCCRCC are evolving, with an increase in combination treatments being approved and new immunotherapies on the horizon. We must remember that RCC is a very heterogeneous tumor and that challenges the identification of biomarkers for this disease [62,63]. We do not know whether liquid biopsy and other emerging molecular technologies could help solve this problem [64]. What is more, it is difficult to isolate markers predictive of treatment response in a fast-changing therapeutic environment. Single-cell sequencing methods, novel PD-L1 tracer-based imaging modalities, ex vivo tumor spheroids for the creation of tumor immunograms, and immuno-PET are some of the most likely translational approaches to predict treatment responses in the immunotherapy era [65,66]. Future directions include next-generation sequencing of circulating tumor DNA and the study of the gut microbiome [67]. Of course, efforts to identify biomarkers evaluating early therapeutic efficacy could be of help to optimize the length of time for effective treatment in each line [68].

ICI targeting the PD-1/PD-L1 interaction and the activation of CTLA-4 via B7-1 or B7-2 are changing the therapeutic landscape in renal cancer. In the CheckMate 214 trial, patients with PD-L1 levels ≥1% before treatment had an ORR of 58% versus 25% after receiving nivoliumab plus ipilimumab versus sunitinib, respectively, and lower levels of PD-L1 expression were correlated with a more favorable risk [67]. However, the real prognostic value of PD-1, PD-L1, and CTLA-4 remains unclear as these biomarkers have been evaluated in clinical trials, but a clear definition of which is the most appropriate cannot be defined at present (Table 3) [67]. 

In metastatic disease, PD-L1 expression in tumor cells or in tumor-infiltrating mononuclear cells (TIMC) has been the most studied biomarker for the prediction of a response to PD-1/PD-L1 checkpoint inhibition therapy [69,70]. Response rates are better in PD-L1 positive tumors, but there is also a significant response in PD-L1 negative ones. Therefore, PD-L1 expression is not a good predictive marker itself, and thereof cannot be used to assign therapy in a particular patient [24,71,72]. Also, the role of CTLA-4 expression in TIMC has been underused in the evaluation of response markers to ICI [72].

Many issues are responsible for failure to develop predictive biomarkers for ICI therapy, including dynamic expression, and the aforementioned heterogeneity within the primary tumor, as well as between primary and metastatic sites. Unfortunately, the pattern of PD-L1 expression differs within areas of the same tumor [6,7], and the identification largely depends on the sampling extent and more precisely on the number of blocks evaluated by immunohistochemistry. A possible explanation for the response to anti-PD-L1 therapy in some patients with PD-L1 negative CCRCC might be inappropriate sampling. PD-L1 immunostaining with monoclonal antibodies recognizing different epitopes also increases the level of uncertainty in the interpretation of the results. Furthermore, the reactivity of different antibodies may also be affected by PD-L1 post-translational modifications [8,73]. Finally, PD-L2 expression either on tumor cells or tumor-infiltrating lymphocytes might partly explain the response to anti-PD-1 therapy in PD-L1-negative CCRCC patients [74]. Another controversial issue that needs to be addressed is the variability in the interpretation of immunohistochemical staining and the evaluation of these findings in daily practice [8].

It is an undeniable paradox that in a disease such as mCCRCC in which all present and future treatment strategies are targeted, a targeted approach for immunotherapy is not currently used [29]. The rationale for the selection of patients that will respond to ICI and those in which treatment resistance could be expected will allow a deeper understanding of ICI at the individual patient level, not only in clinical trials but also in clinical practice. Then, and only then, immunotherapy will make a huge impact on patients with metastatic kidney cancer.

## 6. Conclusions

Under the light of randomized clinical trials ICI is becoming the first-line treatment of mCCRCC. Survival benefit has been demonstrated for pembrolizumab plus axitinib combination for all risk groups and for ipilimumab and nivolumab combination in the intermediate and poor-risk groups. Sunitinib and pazopanib stand as the alterative options for all risk groups and cabozantinib for the intermediate and high-risk group as well. The indication for CN is also changing and its current role should also be investigated under the light of new immunotherapies. Unfortunately, optimal markers of response to ICI have not yet been identified either.

## Figures and Tables

**Figure 1 cancers-11-01227-f001:**
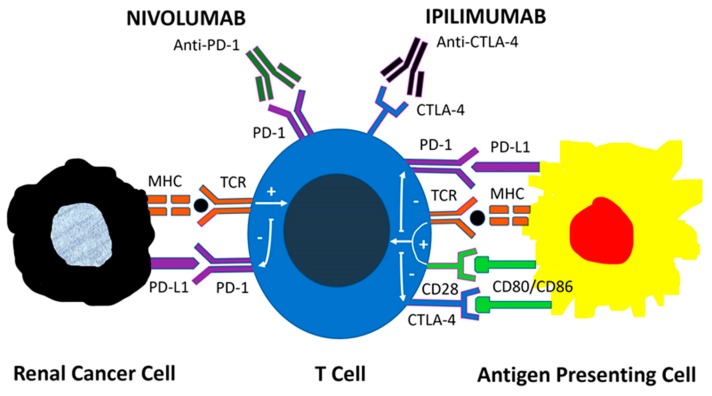
Mechanisms of action of immune checkpoint inhibitors. PD-1 acts as a negative regulator of T-cell activity by binding to PD-L1 on tumor cells and antigen-presenting cells, leading to downstream signaling that inhibits the antitumor T-cell response. CTLA-4 also negatively regulates T-cell activation by binding to B7 ligands CD80 and CD86 on antigen-presenting cells, thus preventing the co-stimulatory interaction between CD28 and B7 ligands. The monoclonal antibodies Nivolumab and Ipilimumab target the immune checkpoint proteins PD-1 and CTLA-4, respectively. Nivolumab blocks the inhibitory signal of the PD1: PD-L1 interaction while Ipilimumab blocks the inhibitory signal of the CTLA-4: B7 interaction.

**Table 1 cancers-11-01227-t001:** Treatment recommendations for first-line and second-line therapy of metastatic clear cell renal cell carcinoma according to the Updated European Association of Urology Guidelines on Renal Cell Carcinoma.

Risk Group/Previous Treatments	Evidence-Based Standard (Level of Evidence)	Alternative Options (Level of Evidence)
IMDC favorable risk	PEMBROLIZUMAB/AXITINIB (1b)	SUNITINIB ^1^ (1b) PAZOPANIB ^1^ (1b)
IMDC intermediate and poor-risk groups	PEMBROLIZUMAB/AXITINIB (1b) IPILIMUMAB/NIVOLUMAB (1b)	CABOZANTINIB (2a) SUNITINIB ^1^ (1b) PAZOPANIB ^1^ (1b)
Second-line prior TKI	NIVOLUMAB (1b) CABOZANTINIB (1b)	AXITINIB ^1^ (2b)
Second-line prior ICI	Any VEGF targeted therapy not previously used in combination with ICI [4]	

^1^ Alternative options with no overall survival benefit proven are specially recommended in patients who cannot tolerate or do not have access to immune checkpoint inhibitors; IMDC: International Metastatic Renal Cell Carcinoma Database Consortium; TKI: Tyrosine kinase inhibitors; ICI: Immune checkpoint inhibitors; VEGF: Vascular endothelial growth factor; Oxford Level of Evidence: 1b (based on at least one randomized controlled phase III trial), 2a (based on at least one randomized controlled phase II trial), 2b (subgroup analysis of a randomized controlled phase III trial), 4 (expert opinion).

**Table 2 cancers-11-01227-t002:** Efficacy results of Phase III clinical trials comparing immune checkpoint inhibitors in combination strategies with single-agent sunitinib.

Combination	Control Arm	Clinical Trial	Primary Endpoints	Results Reported
Ipilimumab + Nivolumab	Sunitinib	CheckMate-214 NCT02231749	ORR, OS, PFS	Intermediate, poor-risk disease: ORR: 42% vs. 29% (*p* < 0.0001) OS: Not reached vs. 26.6 mo (*p* < 0.0001) PFS: 8.2 vs. 8.3 mo (*p* = 0.001) Favorable risk disease: ORR: 39% vs. 50% (*p* = 0.14) OS: Not reached vs. Not reached (*p* = 0.44) PFS: 13.9 vs. 19.9 mo (*p* = 0.189)
Pembrolizumab + Axitinib	Sunitinib	Keynote-426 NCT02853331	OS, PFS	OS: 89.9 vs. 78.3 at 12 mo (*p* < 0.0001) PFS: 15.1 vs. 11.1 mo (*p* < 0.001)
Avelumab + Axitinib	Sunitinib	Javelin Renal-101 NCT02684006	OS, PFS in PD-L1(+)	OS: Not yet reported PFS: 13.8 vs. 7.2 mo (*p* < 0.0001)
Atezolizumab + Becacizumab	Sunitinib	IMmotion-151 NCT02420821	ORR, OS, PFS in PD-L1(+)	ORR: 43% vs. 35% OS: Not yet reported PFS: 11.2 vs. 7.7 mo (*p* = 0.02)
Lenvatinib + Pembrolizumab	Lenvatinib + Everolimus or Sunitinib	Clear NCT02811861	PFS	PFS: Not yet reported
Cabozantinib + Nivolumab	Sunitinib	CheckMate-9ER NCT03141177	PFS	PFS: Not yet reported

ORR: Overall Response Rate; OS: Overall Survival; PFS: Progression-Free Survival; mo: months.

**Table 3 cancers-11-01227-t003:** Protein expression of immunological markers and their clinical significance in clinical trials.

Markers on Immunohistochemistry	Significance
PD-1 Positive in TIMC PD-L1 Positive in Tumor Cells PD-L1 Positive in TIMC CTLA-4 ≥ 2% in TIMC PD-1 in TIMC Positive and CTLA-4 in TIMC ≥2%	Higher grade, OS Histologic variant, high grade Histologic variant OS, CSS Histologic variant, High-grade, High-stage, OS, CSS

TIMC: Tumor-infiltrating mononuclear cells; OS: Overall survival; CSS: Cancer specific survival.

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
