# Peer review of "The Changing Therapeutic Landscape of Metastatic Renal Cancer"

_cancers, 2019, doi:10.3390/cancers11091227_

Round 1
Reviewer 1 Report
Present review article is one more attempt to compile current therapeutic approaches for treating metastatic renal cancer. There are lot of material regarding current therapeutic approaches of metastatic renal cancer especially there is one article entitled 'The Changing Landscape of Management of Metastatic Renal Cell Carcinoma: Current Treatment Options and Future Directions' by Salgia NJ, Dara Y, Bergerot P, Salgia M and Pal SK.Authors should clearly mention unique aspect of the current review.
Author Response
REVIEWER 1
Present review article is one more attempt to compile current therapeutic approaches for treating metastatic renal cancer. There are lot of material regarding current therapeutic approaches of metastatic renal cancer especially there is one article entitled 'The Changing Landscape of Management of Metastatic Renal Cell Carcinoma: Current Treatment Options and Future Directions by Salgia NJ, Dara Y, Bergerot P, Salgia M and Pal SK. Authors should clearly mention unique aspect of the current review.
Response: the reference suggested by reviewer is already included in the article as reference #27 (now 29 after addeing new references, see comment 1 reviewer 2) It has been cited several times, as it is a very recent and important review on the topic. Following the reviewers advice this review is already cited in the last paragraph of the article, to stress the lack of effective biomarkers for immunotherapy.
Reviewer 2 Report
IMMUNE CHECKPOINT INHIBITION IN RENAL CANCER
This mechanism of action discussion has statements without references. For example the first sentence. Please provide references of the statements from lines 37-43
THE NEW PARADIGM TO TREAT METASTATIC RENAL CANCER
“Systemic therapy is the mainstay of treatment with patients with metastatic disease” Does this need to be stated? Line 91 need to say “The last 15 years saw….. Line 94 need to say “a potent VEGF……. Line 115 need to say “the combination of nivolumab…. Do the authors want to discuss papillary tumors as well briefly?CYTOREDUCTIVE NEPHRECTOMY IN THE ERA OF IMMUNOTHERAPY
Line 240, not sure you can say patients would likely benefit from cytoreductive nephrectomy, there is no level 1 data to support this yet. I would temper the statement
THE NEED FOR NEW MARKERS IN THE ERA OF IMMUNOTHERAPY
This discussion needs to be elaborated. There are many more markers besides CTLA and PDL1 that have been included in the above mentioned trials A table summarizing these markers and their ability to predict response would be extremely helpful to the reader.Author Response
REVIEWER 2
IMMUNE CHECKPOINT INHIBITION IN RENAL CANCER
This mechanism of action discussion has statements without references. For example, the first sentence. Please provide references of the statements from lines 37-43
Response: new references (1 and 2) are provided as suggested by reviewer. All references have been renumbered consequently.
THE NEW PARADIGM TO TREAT METASTATIC RENAL CANCER
“Systemic therapy is the mainstay of treatment with patients with metastatic disease” Does this need to be stated?
Response: we consider, at least from the urological point of view this assessment is necessary.
Line 91 need to say “The last 15 years saw…
Response: the correction has been performed.
Line 94 need to say “a potent VEGF…….
Response: the correction has been performed.
Line 115 need to say “the combination of nivolumab….
Response: the correction has been performed.
Do the authors want to discuss papillary tumors as well briefly?
Response: we consider papillary tumours and other histology are not the intention of this review, that focuses on clear cell renal cell carcinoma alone.
CYTOREDUCTIVE NEPHRECTOMY IN THE ERA OF IMMUNOTHERAPY
Line 240, not sure you can say patients would likely benefit from cytoreductive nephrectomy, there is no level 1 data to support this yet. I would temper the statement
Response: we consider the statement is already tempered as the observation is focused exclusively on good risk patients, and not on all metastatic cases. We feel this statement is the proper and balanced consideration after literature review.
THE NEED FOR NEW MARKERS IN THE ERA OF IMMUNOTHERAPY
This discussion needs to be elaborated. There are many more markers besides CTLA and PDL1 that have been included in the above mentioned trials. A table summarizing these markers and their ability to predict response would be extremely helpful to the reader.
Response: this part of the article has been improved following the suggestion of the reviewer. New references (new # 63, 64, 67 and 68) have been included and a table summarizing markers evaluated and their likely value has been added.